# Accelerating network layouts using graph neural networks

Csaba Both[1], Nima Dehmamy[2], Rose Yu[3] & Albert-László Barabási [1,4,5] ✉

Graph layout algorithms used in network visualization represent the first and the most widely used tool to unveil the inner structure and the behavior of complex networks. Current network visualization software relies on the force-directed layout (FDL) algorithm, whose high computational complexity makes the visualization of large real networks computationally prohibitive and traps large graphs into high energy configurations, resulting in hard-to-interpret "hairball" layouts. Here we use Graph Neural Networks (GNN) to accelerate FDL, showing that deep learning can address both limitations of FDL: it offers a 10 to 100 fold improvement in speed while also yielding layouts which are more informative. We analytically derive the speedup offered by GNN, relating it to the number of outliers in the eigenspectrum of the adjacency matrix, predicting that GNNs are particularly effective for networks with communities and local regularities. Finally, we use GNN to generate a three-dimensional layout of the Internet, and introduce additional measures to assess the layout quality and its interpretability, exploring the algorithm's ability to separate communities and the link-length distribution. The novel use of deep neural networks can help accelerate other network-based optimization problems as well, with applications from reaction-diffusion systems to epidemics.

The numerical and analytical toolset of network science has played a key role in the scientific community's ability to explore large complex systems, helping to predict and manage the COVID pandemic[1,2], identify drug repurposing opportunities[3], quantify traffic patterns in cities[4], or understand the spread of fake news[5,6]. The first step of network analysis requires us to visualize the network of interest, a process supported by multiple software packages. The two most popular visualization packages, Cytoscape[7] and Gephi[8], have been used in over 40,000 publications, documenting the wide and cross-disciplinary role of graph layouts from systems biology to ecology, social sciences, and even literature. Yet, most visualization efforts are limited to networks of hundreds, occasionally a few thousand nodes, constrained by the computational complexity of the existing algorithms.

Network visualization relies on different implementations of the force-directed layout (FDL)[9–12], a graph layout algorithm that treats links as springs that pull connected nodes close to each other

and relies on short-range repulsive forces to avoid node overlap. Inspired by energy minimization in computational chemistry[13], the final layout is obtained by minimizing the total potential energy using gradient descent. While widely effective for hundreds of nodes, the $O(N^2)$ computational cost per iteration makes the algorithm prohibitively expensive for larger networks. Hence, our ability to explore large real systems, like the protein-protein interaction network of a human cell with 20,000 proteins and 300,000 links, or networks emerging in social media with millions of nodes, is hindered by computational complexity, placing fundamental limitations in our ability to unveil their architecture. Attempts to visualize the structure of such large systems often result in "hairballs," i.e. high energy layouts that are difficult to interpret and offer only limited insights into the architecture of the network. For this reason, visualizations of very large networks are rarely seen in journals or in the media.

[1]Network Science Institute, Northeastern University, Boston, MA, USA. [2]MIT-IBM Watson AI Lab, IBM Research, Cambridge, MA, USA. [3]Department of Computer Science and Engineering, University of California, San Diego, CA, USA. [4]Department of Medicine, Brigham and Women's Hospital, Harvard Medical School, Boston, MA, USA. [5]Department of Data and Network Science, Central European University, Budapest, Hungary. ✉e-mail: barabasi@gmail.com

## Results

Here we propose an unsupervised machine learning based process to accelerate FDL, demonstrating the potential of deep learning to dramatically speed up graph layout. The key to our approach are Graph Neural Networks (GNN)[14,15], which we use to reparametrize the energy-based optimization problem behind FDL. The resulting NeuLay algorithm is one or two orders of magnitude faster than the existing layout methods, opening up the possibility to quickly and reliably visualize large graphs. Importantly, the algorithm often converges to lower energies than those accessible by FDL, identifying more optimal layouts with clearer and more informative structures. We analytically show that the superior performance of NeuLay is driven by the neural networks' ability to take advantage of the large-scale architecture of the network, resulting in quantifiable and visually apparent differences in the quality of the layout.

### Neural Networks for graph layout

Let $X = (x_1, \ldots x_N)$ be a $N \times d$ matrix that captures the location $x_i$ of node $i$ in $d$-dimensional Euclidean space. FDL performs gradient descent (GD) to minimize the total energy using a loss function $\mathcal{L}(X)$ (see Methods A.), formally written as

$$\frac{dx_i}{dt} = -\varepsilon \frac{\partial \mathcal{L}}{\partial x_i} = -\varepsilon [LX]_i - \varepsilon \frac{\partial V_{NN}}{\partial x_i}, \tag{1}$$

where $\varepsilon$ is the learning rate and $L$ is the graph Laplacian. Computing the $LX$ (elastic forces) term has time complexity $O(N)$ for sparse graphs, and $O(N^2)$ for dense graphs. $V_{NN}$ is repulsive energy helping avoid node overlap, with complexity $O(N^2)$, which can be decreased to $O(N \log N)$ by the Barnes-Hut algorithm[16], hence on dense graphs the bottleneck remains the calculation of the elastic forces (Supplementary Information B Computational Complexity). The core idea of our approach is to represent the node positions $X$ as the output of a neural network, relying on two architectures: (1) NodeMLP, that starts from a high dimensional random embedding of the nodes and finds a map to the target dimension $d = 3$ of the layout; (2) NeuLay, that exploits the graph structure via Graph Convolutional Networks (GCN)[14] (Fig. 1b, c and Methods B). NeuLay is a flexible framework and allows for the use of different GNN architecture other than GCN, such as Graph Attention (GAT)[17] or Graph Network (GN)[18]. Our experiments (Fig. S11) show similar performance when using GCN, GAT or GN, in terms of speedup and final energy. Hence here we focus on GCN in NeuLay due to its simplicity. In NeuLay-2, we apply two GCN layers and then concatenate the layer outputs to obtain a high dimensional node embedding, which is then projected down to $d = 3$ dimensions. In our method, unlike more familiar uses of deep neural networks, retraining of the model is required for each graph layout as the training process is the optimization of the FDL which needs to be performed for every new graph layout. As we show next, the proposed GNN-based method improves computational complexity by reducing the number of iterations required for convergence, rather than reducing the per-step time complexity.

### NeuLay offers more optimal layouts faster

To assess performance, we rely on two figures of merit: speed and quality. For speed, we examine the running time ('wall-clock' time). As a proxy for the layout quality, we explore several measures. The most natural one is the potential energy (loss value) of the final layout which we find to strongly correlate with the quality of the layout. But we also explore two additional measures, such as cluster separation and link length distribution. We begin by comparing the performance of FDL with the three proposed neural network models, NodeMLP, NeuLay, and NeuLay-2 for a simple cubic lattice (Fig. 1d, and Fig. S1b). We find that while NodeMLP and NeuLay offer significant speedup in laying out this network with a known optimal layout (Fig. 1f), NeuLay-2 with two

GCN layers has the fastest convergence of the energy, prompting us to focus on this architecture hereafter. Furthermore, we measured the speedup using GPU hardware (see in the Fig. S11), consistently observing results similar to that reported in Fig. 1g, h.

We compared the speedup for four networks constructed using various graph generation models, like the Erdős-Rényi (ER) random graph algorithm, Barabási-Albert[19] (BA) model, Stochastic Block Model (SBM)[20], and Random Geometric Graphs (RGG)[21]. While these networks span drastically different topologies, sizes, and link densities, in all cases NeuLay-2 reaches the final state one to two orders of magnitude faster than FDL (Fig. 1h, i). We find that the speedup increases with the number of nodes and links (Fig. 1h, i), and falls with increasing network density (Fig. 1j). The speedup is particularly remarkable for graphs with a strong community structure, such as networks generated by the stochastic block model (SBM), and grid-like graphs, like the random geometric graph (RGG) (red symbols in Fig. 1h–j), compared to graphs that lack local structure, like the ER and BA networks (blue and green symbols in Fig. 1h–j). Yet, we observe speedup for each of those networks for a fixed density, finding that the speedup scales as $N^{0.8}$ for 2D RGG, $N^{0.3}$ for BA networks, and $N^{0.2}$ for ER random graphs (Fig. 1h).

NeuLay-2 is not only faster, but also identifies better layouts. Indeed, while for small and simpler networks, like the cubic lattice (Fig. 1d), FDL and NeuLay-2 converge to indistinguishable energies, for larger networks NeuLay-2 identifies a deeper energy minimum compared to FDL (Fig. 1g). To systematically quantify this difference, we measured the ratio between the final energy of FDL and NeuLay-2 ($\Delta E = E_{FDL}/E_{NeuLay-2}$). We find this ratio to increase with the size of the network (Fig. 1g), indicating that for large networks FDL gets trapped into a local sub-optimal configuration, successfully avoided by NeuLay-2. This ratio is especially large and increasing with $N$ for BA and ER graphs, indicating that while NeuLay-2 may not show as high speedup over FDL for these networks as it does for more structured architectures, like RGG, it offers a significant advantage in terms of energy. As we show later, this energy difference has a dramatic impact on the quality of the final layout.

### Large structures and outlier Eigenvalues help accelerate the layout

The higher speedups observed for networks generated by SBM and RGG, characterized by communities (SBM) and spatial proximity (RGG), suggests that the speedup is related to the leading eigenvalues of the adjacency matrix. To test this hypothesis, we analytically derived the speedup, finding that: (i) Speedup of NeuLay-2 is expected to increase with of the number of outlier eigenvalues; (ii) As a falsifiable test, we predict that removing the outlier eigenvalues should significantly reduce the speedup of NeuLay-2; (iii) Keeping only the outlier eigenvalues should be sufficient to achieve a speedup comparable to using the full spectrum.

We tested predictions (i)–(iii) on networks generated by SBM, for which the number of outlier eigenvalues equals the number of blocks (communities), allowing for direct control of the spectrum. Figure 2a shows the evolution of FDL vs NeuLay-2 for four communities, indicating that in NeuLay-2 the communities converge to their final positions by step 100, much earlier than in FDL. We find the speedup for SBM to grow with the number of blocks (communities) as $\sim n_{block}^{0.77}$ (Fig. 2b). Plotting the speedup vs number of outlier eigenvalues, (Fig. 2c), we find that for SBM (as well as for RGG), the speedup increases as $\sim n_{out}^{0.96}$ with the number of outliers, validating prediction (i). Yet, it is not clear if the correlation between the speedup and the number of outliers is causal, or it is driven by some other uncontrolled features of the graphs. If the outlier eigenvalues are truly responsible for the speedup, replacing $A$ with a similar matrix that lacks the outliers must reduce the speedup. We, therefore, used the spectral expansion, $A = A_{top} + A_{bulk}$, to separate the outliers, ($A_{top}$, Fig. 2e, red part of the histograms) and the rest of the modes ($A_{bulk}$, Fig. 2e, blue part of the

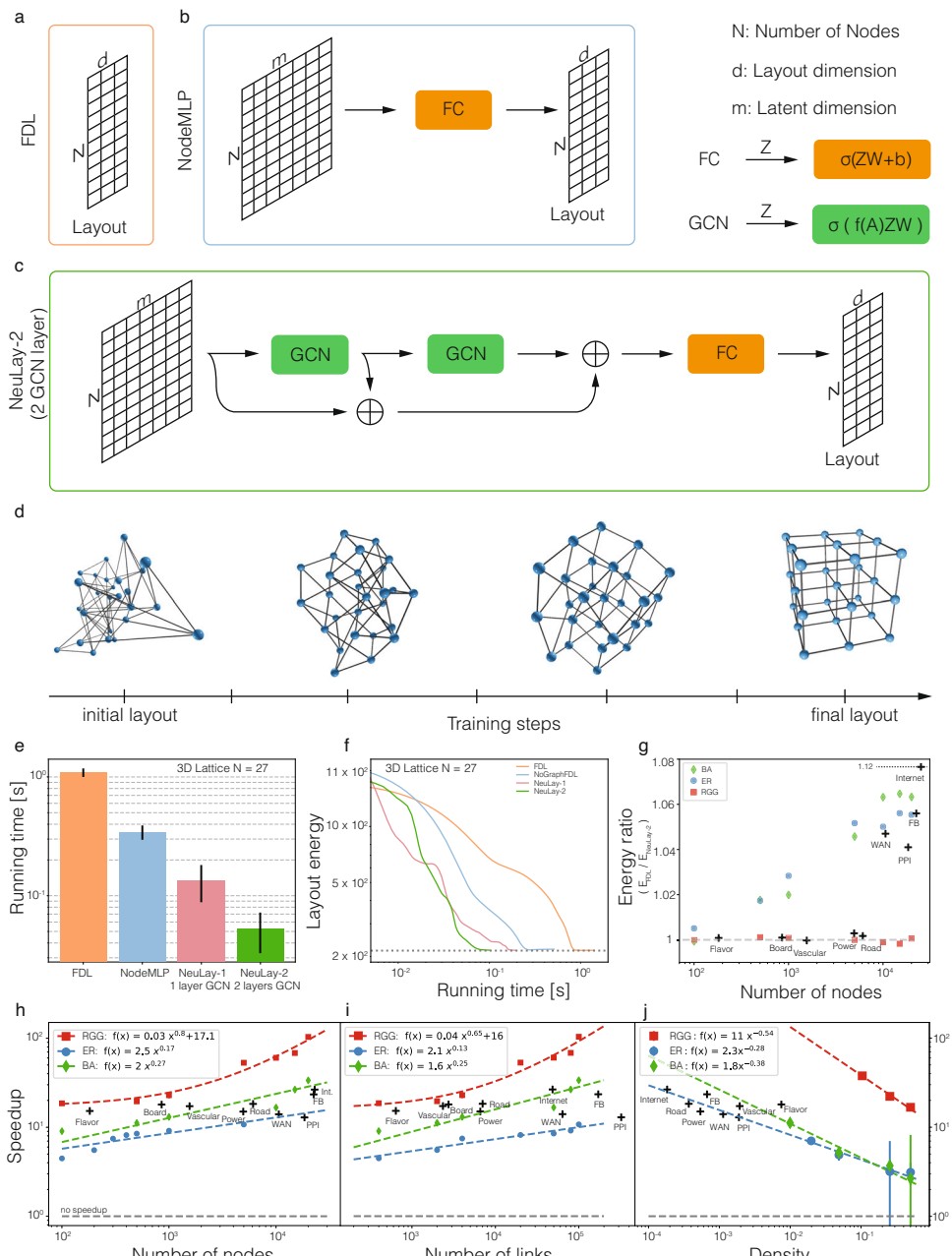

**Fig. 1 | Laying out networks using neural network. a** FDL optimizes the $d$ dimensional node positions to find a network layout. **b** NodeMLP replaces the $d$ dimensional input by a neural network that relies on a fully connected layer (*FC*) to project the high dimensional embedding to the $d$ dimensional layout. **c** NeuLay encodes the graph structure by graph neural networks, (*GCN*), that maps the adjacency matrix to the node positions. We find that for large networks, two *GCN* layers are optimal, as more than two layers can slow down the computation, while a single layer does not offer the highest speedup. **d** The evolution of a simple cubic lattice, starting from a random configuration, showing its gradual convergence to the lowest energy state as the FDL algorithm identifies its layout. **e** The running time of the four tested models for a cubic lattice with 27 nodes. **f** The NeuLay-2 (green) achieves the same final energy state as all other models but converges faster. **g** The energy ratio of NeuLay-2 and FDL for networks generated by different models (BA, ER, and RGG) indicates that FDL becomes trapped in higher energy local minima. The energy ratio increases with network size. The dependence of speedup (the running time (`wall-clock' time) ratio of FDL and NeuLay models) in function of (**h**) the number of nodes $N$ and (**i**) the number of links $L$ in the network, for fixed density graphs. **j** Keeping fixed the number of nodes we find that the speedup decreases with density, $L/N$. The gray lines corresponds to no speed up, the blue line (green, red) is the speedup for the ER (BA, RGG) network, respectively. We also measured the speedup for several real networks, like the Flavor network[26], Norwegian Boards of Directors (public companies)[27], Mouse vascular network[28], US Power Grid[29], Word Association Network[22], Road network-Oakland[30], Protein-protein interaction network[23], Facebook social network[24] and the network of Internet at the level of autonomous system[25].

histograms, also see Methods C.). We find that for SBM and RGG, networks with multiple outliers, using only the outlier eigenvalues $A_{top}$ results in higher speedup than using the full spectrum $A$, in line with prediction (iii) (Fig. 2d, red bars). In contrast, removing the outliers of the RGG spectrum and using only $A_{bulk}$ in NeuLay-2 dramatically

reduces the speedup (Fig. 2d, blue bars), supporting prediction (ii). Finally, in line with the prediction (i), we do not observe a difference in the speedup by using $A$, $A_{top}$, or $A_{bulk}$ in networks that lack outlier eigenvalues, like networks generated by the ER and the BA model (Fig. 2d).

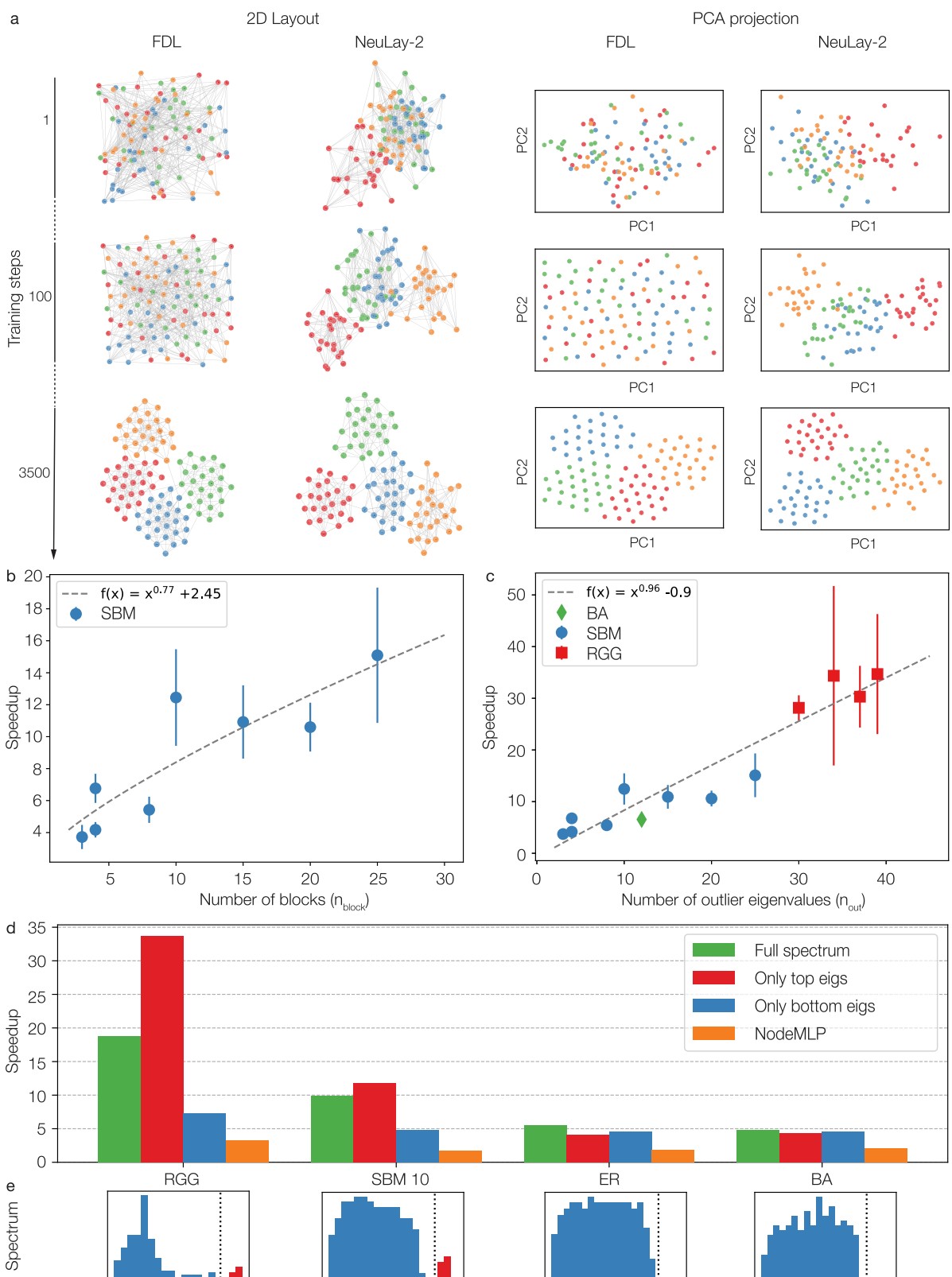

**Fig. 2 | The role of leading eigenvalues in network layout. a** Comparing the training steps of FDL (left column) and NeuLay-2 (right column) on a SBM graph with 100 nodes and four 25 node blocks. The PCA projection, showing in the right two columns, is colored for the blocks in the graphs. As the panels show, NeuLay-2 separates the blocks early, in contrast with FDL that finds the blocks only at the very end. **b** Speedup in the function of the number of blocks. **c** Speedup in the function

of the number of outlier eigenvalues that separate from the Wigner semicircle, indicating that the higher number of outlier eigenvalues yield higher speedup. **d** The NeuLay-2 performance using three different graph eigenvalue decompositions in GCN modules. **e** The spectrum line illustrates the separation if eigenvalues included in **d**.

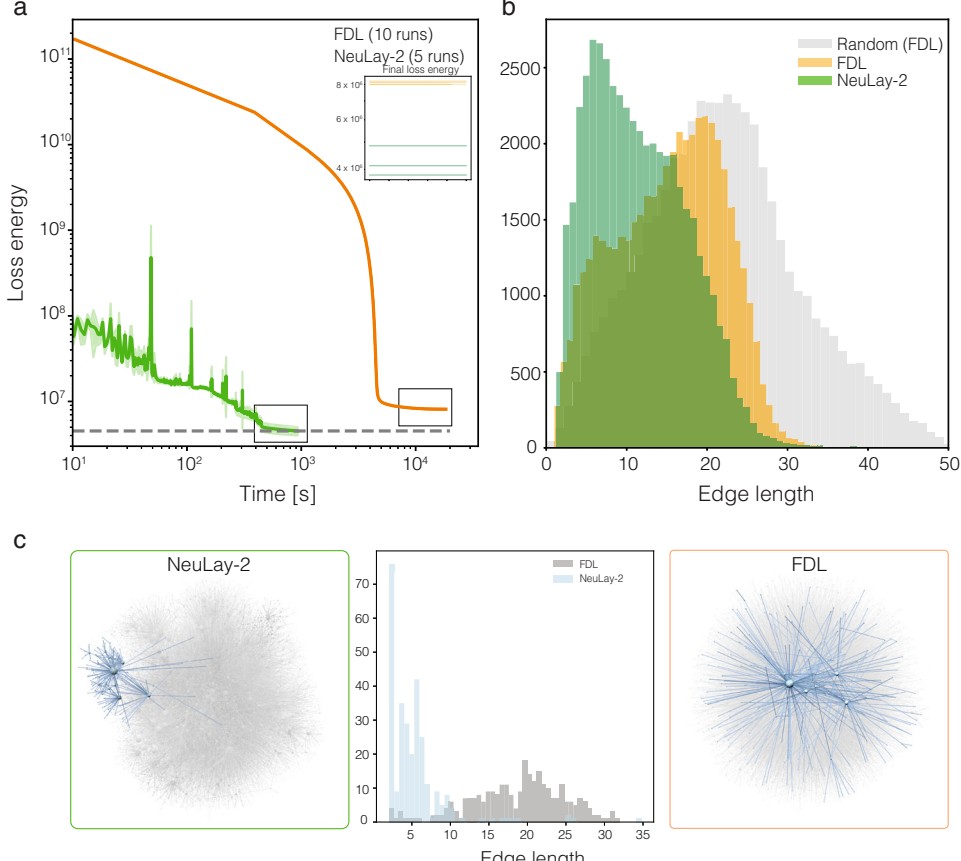

**Fig. 3 | The interpretability of the layout.** The Internet, with 22, 963 nodes, and 48, 436 links is large enough to represent a difficult visualization task for the existing algorithms. **a** The time-energy plot shows that NeuLay-2 converges faster and finds a considerably lower energy state. NeuLay-2 reparameterizes node positions from the initial iteration, resulting in a significant decrease in the loss-energy. Indeed, FDL gets trapped in a higher energy state and is unable to reach the NeuLay-2 energy level even after a much longer time. The curves correspond to ten independent runs, that converge to slightly different final energies, as shown in the inset. **b** The link length distribution in the layouts, confirming that the FDL layout has more long links compared to the NeuLay-2 layout. It also reveals that both layouts differ from a geometrically randomized layout. **c** The histogram in the middle panel shows the link length distribution for FDL and NeuLay-2 for a community identified by the Louvain algorithm. On the left side of the histogram we show the NeuLay-2 layout, while on the right side the FDL layout, highlighting with blue nodes and links the same community. See Fig. S10 for the other communities, that display the same pattern.

Note that in most network visualization problems we do not know the relevant eigenvalues, nor the eigenvalue combination that offers the best optimization. Yet, NeuLay-2 automatically identifies structures useful for accelerating FDL, and offers a fast convergence whether the network is dominated by outliers (like SBM and RGG), or lacks multiple outliers (BA, ER), hence improving the layout of arbitrary networks.

## The quality of large network layouts

To illustrate the practical value of NeuLay-2, we used it to lay out multiple large real networks in three dimensions, $d = 3$, like the word association graph (WAN)[22] ($N = 10,617$, $L = 63,781$), the human protein-protein interaction network (PPI)[23] ($N = 18,448$, $L = 322,285$), Facebook social network[24] ($N = 22,470$, $L = 171,002$), and the Internet at the autonomous system level[25] ($N = 22,963$, $L = 48,436$). For comparison, we laid out multiple smaller networks as well, like the flavor network[26] ($N = 182$, L = 641), boards of directors (public companies in Norway)[27] ($N = 854$, $L = 2745$), mouse vascular network[28] ($N = 1558$, $L = 2352$), US power grid[29] ($N = 4941$, L = 6594), and the road network in Oakland[30] ($N = 6105$, $L = 7,029$) (Fig. 1f–h). In all cases, we find NeuLay-2 to be an order of a magnitude faster than FDL, resulting in a 14-fold improvement in speed for WAN and a 13-fold improvement for PPI (Fig. 1f–h). Even more important is the fact that for each real network NeuLay-2 converges to a deeper energy state, a difference that is particularly

remarkable for large networks, like the PPI and WAN. We observe the most dramatic improvement in the case of the Internet, for which previous successful visualization efforts had to reduce the network to its backbone[31]. Indeed, we find that FDL becomes trapped in a sub-optimal layout, whose energy is 12% larger than the one identified by NeuLay-2 (Fig. 3a, and Fig. 4a, c). To ensure that this sub-optimal configuration is not a result of an accidental trapping of FDL in some local minima, we have re-run both NeuLay-2 and FDL ten times, starting from different initial configurations, each time observing largely indistinguishable time and energy curves (Fig. 3a).

The lower energy identified by NeuLay-2 has a visually detectable impact on how informative the layout is: while the higher energy NeuLay-2 layout has an observable local community structure (Fig. 4a, b), the FDL layout appears to be largely random (Fig. 4c, d), reminiscent of an unstructured hairball. To better assess how well the two layouts capture the inherent structure of the network, we used the Louvain algorithm[32] to identify 36 communities in the Internet graph, coloring 12 of them on Fig. 4 for visual clarity. As Fig. 4a, c indicate (see also the video https://vimeo.com/732791412), while in the NeuLay-2 layout nodes in the same community are spatially co-localized, the FDL distributes the community members throughout the layout, failing to co-localize them. To quantify this difference, we measured the link length distribution of each community's internal links (Fig. 3c, and Fig. S10), finding that the distribution identified by NeuLay-2 is much

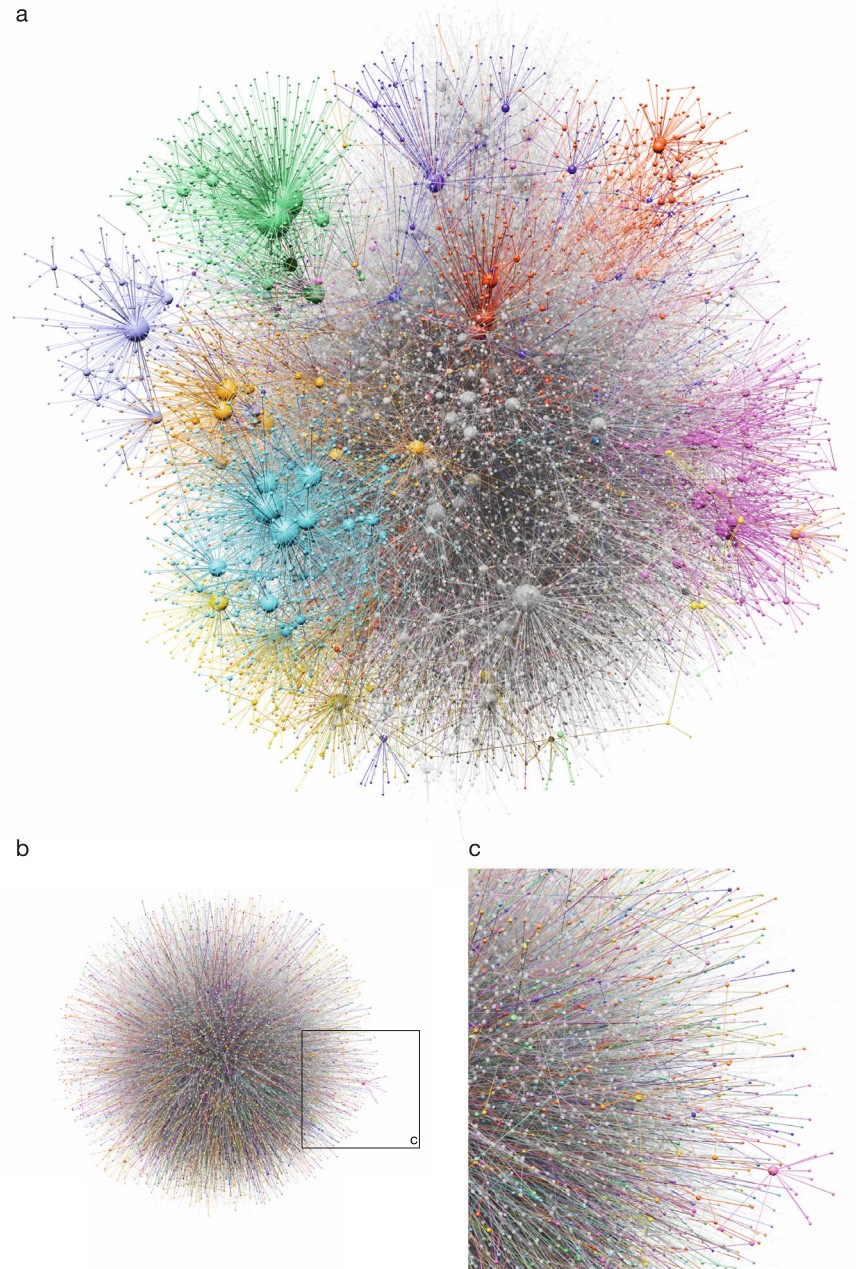

**Fig. 4 | The 3D layout of the Internet. a** The layout generated by NeuLay-2. We used the Louvain algorithm[32] to identify the community structure of the network and for visual clarity we highlight 12 communities in color. For a better view of the full 3D representation, see the video https://vimeo.com/732791412. **b** The FDL, by being trapped in a higher energy state, fails to identify the local communities, and the large scale layout appears to be random, resembling a "hairball". **c** A local zoom into the FDL layout, documenting the absence of community structure.

narrower than the one identified by FDL, confirming better spatial localization. These local differences also impact the global link length distribution of the two layouts (Fig. 3b), indicating that the FDL layout generates more long links than the NeuLay-2 layout, which also explains its larger elastic energy. Additionally, we have introduced a spatial similarity metric measuring how well the clusters are separated in the final layout compared to the FDL layout, finding that NeuLay-2 not only discovers but also better separates the clusters in the final layout (Fig. S8).

The higher energy state to which FDL converges does not necessarily result in a random layout. To see this, we apply a geometric randomization, by randomly exchanging the nodes, while keeping the physical coordinates of the layout and the adjacency matrix unchanged. We find that the link length distribution in the FDL layout is

shorter than expected under geometric randomization (Fig. 3b), indicating that FDL does converge to a non-random low energy layout. Yet, its higher energy compared to the layout identified by NeuLay-2 results in FDL's failure to identify the network's inherent local community structure.

## Discussion

The proposed NeuLay algorithm, a Graph Neural Network (GNN) developed to parameterize node features, significantly improves both the speed and the quality of graph layouts, opening up the possibility to quickly and reliably visualize large networks. It offers a fast and easy-to-use tool for large network visualization. We find that, many large networks have informative large-scale structures that remain hidden if the layout algorithms do not extract their main structural

characteristics and find a way to display them. As we have shown here, NeuLay excels at this task, producing a high-quality layout, with distinct clusters and a clear internal structure. It achieves this performance by speeding up the dynamics of slow modes. Indeed, the leading eigenvectors of the adjacency matrix, or Principal Components (PC) in machine learning, are the "slow modes" in the dynamics of FDL[33,34]. NeuLay projects the graph layout to the top few PC (Fig. 2a) from the first iteration, separating the large communities which slows the dynamics, and catalyzing a faster convergence.

The mechanism applied by NeuLay to accelerate convergence is not restricted to graph layouts, but can be applied to any energy minimization problem on graphs, or graph dynamical processes expressed as gradient descent. Indeed, FDL is a special case of general reaction-diffusion problems on graphs, where in (1) $LX$ is the "diffusion" and $F_{NN} \equiv -\partial V_{NN}/\partial x$ are the nonlinear "reaction" terms. As our theoretical results do not depend on the exact form of $V_{NN}$, they apply to any problem in the reaction-diffusion class, independent if the node features $x_i$, are densities (e.g. of material flowing on the graph), or probabilities (e.g. susceptible, or infected nodes in epidemic spreading). Hence the method can improve the finding of endemic state in epidemics[35], help with interventions and mitigation[36], improve the modeling of cascading failures[37], and help find optimal graph layout in chip design[38], as well as accelerate models capturing opinion dynamics in social media[6,39].

Currently, the efficiency of NeuLay is limited only by the computational complexity of GNN, which, while considerably faster than FDL, can still be expensive on exceptionally large graphs. We foresee further improvement by exploiting symmetries or hierarchical structures[40] present in networks, leading to more efficient message-passing in GNN. These ideas could result in more advanced GNN architectures similar to GraphSage[41] and ClusterGCN[42], which make the graph sparser and thus reduces the computational complexity of GNN. It would be equally valuable the development of GNN or other AI-based tools to accelerate the layout of physical networks whose links are not straight, but curve to avoid overlaps[43,44], capturing network layouts observed in the brain connectome or metamaterials.

## Methods

### Force Directed Layout (FDL)

Consider an undirected network with $N$ nodes and $A \in \mathbb{R}^{N \times N}$ adjacency matrix, where $A_{ij}$ is the weight of the link connecting node, $i$ and $j$, and denote with $X = (x_1, \ldots x_N)$ the $N \times d$ matrix that captures the location $x_i$ of node $i$ in a $d$-dimensional Euclidean space. FDL brings connected nodes close by minimizing the total energy, $\mathcal{L}$, that also plays the role of the "loss function" in machine learning[9–12]:

$$\mathcal{L}(X) = V_{el} + V_{NN}, \qquad V_{el} = \frac{1}{2}\sum_{i,j} A_{ij}|x_i - x_j|^2 = \frac{1}{2}\text{Tr}[X^T L X] \qquad (2)$$

where $L = D - A$ is the graph Laplacian and $D_{ij} = \delta_{ij}\sum_k A_{ik}$ is the degree matrix. For the repulsive potential $V_{NN}$ we choose a short-range Gaussian repulsion $V_{NN}(X) = a_N \sum_{ij} \exp(|x_i - x_j|^2/4r_0^2)$[43], but any rapidly falling repulsive potential would work. FDL performs gradient descent (GD) to minimize the total energy (eq.(1)). Note that in FDL the repulsive potential is usually chosen to be "long-range", e.g. $V_{NN} = a_N \sum_{ij} r_0/\|x_i - x_j\|$. This results in an all-to-all repulsive force with complexity $O(N^2)$. The Barnes-Hut algorithm[16] can be used to reduce this to $O(N \log N)$. Despite the widespread use of long-range repulsive forces for the layout of large and sparse graphs, short-range repulsive forces are lower complexity (O(N)). We note that, while FDL with short-range forces failed to yield a good layout for the Internet graph, FDL using long-range forces does yield a good layout. However, because long-range forces can become intractable for large graphs, we implement short-range forces[43]. Our experiments on the Internet graph show that NeuLay does not require long-range forces to find good layouts for large graphs.

### Reparametrizing node positions with deep neural networks

To reparametrize $X$, we introduce two architectures: NodeMLP and NeuLay, described in Figure 1 and in Supplementary Information A. NodeMLP starts from an $N \times h$ dimensional random $Z$ embedding of the nodes. It then projects to the target dimension $d = 3$ of the layout by defining node positions as $X = \sigma(ZW + b)$, where $\sigma$ is a nonlinear function such as $tanh$ and $\theta = \{Z, W, b\}$ are the set of trainable parameters of the neural network. NeuLay uses GNN, that starts from an $N \times h$ random embedding $Z$ and applies a Graph Convolutional Networks (GCN)[14] layer to obtain $G_1 = \sigma(f(A)ZW^{(1)})$, with $f(A) = \tilde{D}^{-1/2}\tilde{A}\tilde{D}^{-1/2}$, where $\tilde{A} = A + I$ and $\tilde{D}_{ii} = \sum_j \tilde{A}_{ij}$ is the degree matrix of $\tilde{A}$. Here $G_1$ is $N \times h_1$ and is a new embedding of nodes in $h_1$ dimensions that incorporates the graph structure via $f(A)$. In the two-layer NeuLay-2, we apply another GCN with output $G_2 = \sigma(f(A)G_1W^{(2)})$ and dimensions $N \times h_2$. Then, we concatenate the layer outputs $G_1$ and $G_2$ with $Z$ along the embedding dimensions to obtain the $(h + h_1 + h_2)$ dimensional node embedding $G = [Z|G_1|G_2]$. Finally, we project $G$ down to $d$ dimension as $X = \sigma(GW + b)$. The set of trainable parameters of NeuLay-2 are $\theta = \{Z, W^{(1)}, W^{(2)}, W, b\}$.

To obtain a layout we input $X(\theta)$ into the FDL algorithm and using the loss function (2). We perform energy minimization using gradient descent. Instead of optimizing $X$ directly, we optimize the neural network parameters $\theta$. Using the chain rule we can rewrite the GD equation (1) in terms of $\theta_a$,

$$\frac{d\theta_a}{dt} = -\varepsilon \frac{\partial \mathcal{L}}{\partial \theta_a} = -\varepsilon \sum_i \frac{\partial x_i}{\partial \theta_a}\frac{\partial \mathcal{L}}{\partial x_i} \qquad (3)$$

Unlike the familiar uses of deep neural networks, where training is done only once, here we retrain each $\theta$ for each layout.

### The role of outliers in the eigenspectrum

To understand the mechanism that drive the faster convergence of NeuLay, we study the spectral expansions $A = \sum_i \lambda_i \psi_i \psi_i^T$ and $L = D - A = \sum_i l_i \phi_i \phi_i^T$. While in general the eigenvectors $\psi_i$ of $A$ and $\phi_i$ of $L$ differ, in RGG and SBM all node degrees are close to an average degree $\langle k \rangle$ and we have $L \approx \langle k \rangle I - A$, yielding $\phi_i \approx \psi_i$ and $l_i \approx \langle k \rangle - \lambda_i$. Therefore, the elastic energy $V_{el} = \text{Tr}[X^T L X]/2$ in (2) dominates in the early stages of the optimization. Using (1) and $\mathcal{L} \approx V_{el}$ to examine the early stage evolution of the overlap of $X$ with $\psi_i$ for graphs where $L \approx \langle k \rangle I - A$, finding

$$\frac{d(\psi_i^T X)}{dt} \approx -\varepsilon \psi_i^T L X = -\varepsilon(\langle k \rangle - \lambda_i)\psi_i^T X. \qquad (4)$$

Hence, for FDL on lattices, RGG, and SBM, the mode $\psi_i^T X$ for each $i$ evolves almost independently of other modes $j \neq i$. Eq. (4) predicts that in early iterations the magnitude of the mode $\psi_i^T X$ drops exponentially with a rate $-\varepsilon(\langle k \rangle - \lambda_i)$, and that modes with the largest $\lambda_i$ drop at the slowest rate during GD (as $L$ is positive semi-definite, $\lambda_i \leq \langle k \rangle$). Importantly, if the spectrum of $A$ contains "outlier" eigenvalues $\{\lambda_o\}$, with $\lambda_o \gg \text{mean}_i[\lambda_i]$, the corresponding modes $\psi_o^T X$ evolve the slowest. Specifically, define the set of outlier eigenvalues as the indices $Out = \{j|\tilde{\lambda}_j > \text{mean}_i[\tilde{\lambda}_i] + \sigma_{\tilde{\lambda}}\}$, with $\sigma_{\tilde{\lambda}}$ being the standard deviation of the eigenvalues. The projection onto top eigenvectors is defined as $A_{top} = \sum_{i \in Out} \tilde{\lambda}_i \psi_i \psi_i^T$ and the rest is $A_{bulk} = A - A_{top}$.

Both NodeMLP and NeuLay start from a random node embedding $Z$. The difference is that NodeMLP performs $X = \sigma(ZW + b)$, while NeuLay applies $G_1 = \sigma(f(A)ZW^{(1)})$. In NeuLay we choose $f(A) = \tilde{D}^{-1/2}\tilde{A}\tilde{D}^{-1/2}$, which for lattices, RGG and SBM, again has approximately the same eigenvectors as $A$. Using the spectrum $f(A) = \sum_i \tilde{\lambda}_i \psi_i \psi^T$ to expand $Z = \sum_i z_i \psi_i$, we find $f(A)Z = \sum_i \tilde{\lambda}_i z_i \psi_i$. In graphs with many outliers significantly larger than the bulk of the eigenvalues, the outliers dominate the spectral expansion and $f(A)Z \approx \sum_{i \in Out} \tilde{\lambda}_i z_i \psi_i$. Hence, when performing GD $dZ/dt = -\varepsilon \partial_Z \mathcal{L}$, in

the presence of a GCN layer the gradients for outliers are magnified by $\tilde{\lambda}_i$, supporting prediction (i), that the more outliers eigenvalues the graph has, the higher the speedup. We used *Out*, the set of outlier eigenvalues to build $A_{top}$ and $A_{bulk}$ to separate the relevance of the outlier eigenvalues in predictions (ii) and (iii). The details for how the outliers eigenvectors result in a faster drop in loss $d\mathcal{L}/dt$ refer to Supplementary Information C.

## Data availability

All data that support the plots within this paper and other findings of this study are available at https://github.com/csabath95/NeuLay.git and the listed public sources: the word association graph (WAN)[22]: [http://w3.usf.edu/FreeAssociation/], the human protein-protein interaction network (PPI)[23], the Facebook social network data[24]: http://snap.stanford.edu/index.html, the Internet at the autonomous system level[25]: http://www-personal.umich.edu/~mejn/netdata/, the flavor network[26], boards of directors (public companies in Norway)[27]: https://networks.skewed.de/net/board_directors, US power grid[29]: http://www-personal.umich.edu/~mejn/netdata/, and the road network in Oakland[30]: http://snap.stanford.edu/index.html.

## Code availability

Code is available for this paper at https://github.com/csabath95/NeuLay.git. All other codes that support the plots within this paper and other findings of this study are available from the corresponding author upon request.

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

## Acknowledgements

We thank A. Grishchenko for help with the 3D visualizations. A.-L.B. and C.B. were supported by ERC grant No. 810115-DYNASET, The Eric and Wendy Schmidt Fund for Strategic Innovation G-22-63228, NSF SES-2219575, and John Templeton Foundation #62452. R.Y. acknowledges support in part by the U.S. Department Of Energy, Office of Science, U. S. Army Research Office under Grant W911NF-20-1-0334, Google Faculty Award, Amazon Research Award, and NSF Grants #2134274, #2107256 and #2134178.

## Author contributions

C.B. developed, ran, and analyzed the simulations, generated figures, and contributed to writing the manuscript. N.D. performed the mathematical modeling and derivations and contributed to writing the manuscript. R.Y. have guided designing the machine learning models and wrote the manuscript. A.-L.B. contributed to the conceptual design of the study and was the lead writer of the manuscript.

## Competing interests

A.-L.B. is the founder of Naring and Scipher Medicine, companies exploring the role of networks in health. C.B., N.D., and R.Y. declare no competing interest.
