## [Peer Review File · Nature Communications]

Accelerating Network Layouts Using Graph Neural NetworksREVIEWER COMMENTS

Reviewer #1 (Remarks to the Author):

I have reviewed the manuscript entitled "Accelerating Network Layouts Using Graph Neural Networks," written by Csaba Both et al. This paper studies a very interesting and important question: how to use graph neural networks to accelerate network visualization layout. This is timely work. We have more data about large-scale networks than ever before. However, we can only visualize a part of the original network using different sampling methods, preventing us from capturing the whole picture of the entire network. In this work, the authors show that deep learning can address both limitations of the FDL algorithm. Overall, the paper is well written. Yet, I do have several concerns and comments on the current form of the manuscript regarding the models, methods, and results of this work. I am pretty open to looking at a revised version if the author can address in a satisfactory fashion the issues discussed below:

1. It is unclear what are the meanings of the symbols in fig 1 g h I j. One can use different function forms for the same data points to fit. Why did the authors choose these?
2. figure 3a indicates that the loss energy of FDL is much larger than NeuLay when Time is small. It seems that the initial loss energy of FDL is very large. If they start from similar initial values, maybe their performances are identical. Furthermore, the curve of NeuLay is less stable than FDL. Here the authors only show 5 runs. Perhaps it is better to show the results for more runs and include the error bar.
3. The authors focus more on the runtime of the approach but do not discuss the performance enough. Therefore, although we observe some useful visualizations from the illustrative figures (figures 1d and 2a), I am still unsure about the "performance." Otherwise, it is unclear whether the visualization is similar to or even better than the approaches you compared.

Reviewer #2 (Remarks to the Author):

The paper presents a new computational approach to generating network layouts. Instead of using, say, spring-like layout algorithms, the paper suggests a machine learning method based on the use of graph neural networks. The main goal is to speed up the computation of the layout structure. A secondary goal is to improve the quality of the layout. The paper is well presented, and with good illustrations of how this would work in large networks.

The concerns are:

1. I am not sure that the main limitation of existing network layout algorithms is the computation cost for large networks. Network layouts are useful for visualization (e.g., to generate an image of the network that can help interpretation of specific results). These images are most useful when the network is small enough for the individual nodes and links to be visible. For very large networks, the network layout is less informative. The computational cost of existing methods is still reasonably low. Force-directed layout algorithms, which have a cost of N^2 , can be used in quite large networks.
2. Given the above, the main contribution of this paper might be the possibility of a GNN algorithm that can generate better (rather than faster) layouts. This is briefly discussed in the paper, but I wish this could be further developed and really made into the main point (to me, the computation time comes across as a secondary benefit).

3. A problem with GNN (and any machine learning approach) is that the result will depend on the model training. It is not entirely clear the best way to do that here. Some in-depth discussion seems necessary.

4. Another thing that is not sufficiently clear is the loss function being optimized. Minimizing energies is a natural choice in traditional layout algorithms based on forces, but is that really the best function to optimize in a machine learning implementation? But if the same function is used, then improvements in layout quality will be limited by that choice.

5. More broadly, if we agree that energy is not necessarily the best function, how to compare the quality of the layouts generated using the GNN algorithm with the layouts generated using FDL algorithms?

6. Fig 4 might be the best visual comparison between the methods. By looking at the figure only, it is not immediately clear the advantage given by the new method in terms of layout quality. There might be a way to improve that.

Minor:

- why is it said that the DeepFDL architecture maps to the target dimension $d = 3$? Isn't the layout in 2D?
- how do the authors plan to make the codes available to the readers?

Despite the concerns above, I consider this a very good paper. Therefore, I fully hope that the authors will be able to address the points above so that a suitably revised version of the paper can be published in Nature Communications.

Reviewer #3 (Remarks to the Author):

In this paper, the authors tackle the highly important task of network visualisation. Being able to visualise graphs well is a critical open problem, which usually forms an integral part of any exploratory phase of network science projects. The quality of the visualisation can often make a significant impact on the salient patterns detected in the data. Hence it is an important and timely problem to solve, certainly within scope for Nature Comms.

The authors propose to plug a graph neural network inside the pipeline that produces node 3D coordinates, which are then optimised using a force-field method (energy minimisation with gradient descent).

On the positive side, this work shows clear quantitative and qualitative outperformance compared to the previous state of the art, demonstrating that GNNs can be an excellent tool in the network layout pipeline. I also appreciated the analysis on how this effect likely comes from modelling outlier eigenvalues of the graph Laplacian. Generally, I find the idea to be sound and to make sense, and it produces network layouts of clearly superior quality, especially for large-scale networks of real-world interest. I particularly liked Figure 4 in this regard -- really beautiful!

However, I do have several questions and concerns which would need to all be clearly addressed before I would be comfortable recommending this work for acceptance. I provide my points here, in rough order of significance.

Q1. The evaluation of the paper is described as follows: "We rely on two figures of merit to assess performance: the running time (number of iterations), that captures the computational cost, and the potential energy (loss value) of the final layout, that measures the quality and ultimately the interpretability of the layout." In my opinion, there are two concerning / unclear aspects with this description.

a) First, 'running time' typically refers to 'wall-clock time' (in seconds), but the paragraph implies that we're actually counting the number of gradient descent steps. I think making this distinction clearer is quite necessary. Further, for a downstream practitioner, I think wall-clock time, and/or the hardware setup, might also be quite meaningful to report -- especially because the various baselines may benefit differently from GPU accelerators, and practitioners may appreciate these considerations.

b) Secondly, without thorough user studies, I would be *very* careful claiming that the layout's energy is a proxy for its interpretability. Surely, the produced examples in the paper are good in this regard, but I would tone down the certainty of the claim that the energy is a measure of interpretability.

Q2. Both the DeepFDL and NeuLay methods start with an embedding vector in each of the nodes, Z . However, after several passes through the paper, I found no clear description in the paper on where this Z comes from. My assumption is that it could be a standard structural node embedding method (e.g. DeepWalk, or the lowest h Laplacian eigenvectors). In either case, it is critical to describe where Z could come from, and also to discuss the computational complexity of obtaining such Z (since, e.g., computing Laplacian eigenvectors may be intractable for too large graphs). On a related note, the authors may want to consider the use of affinity-based measures (e.g. effective resistance or hitting time) for powerful embeddings. In the recent preprint "Affinity-Aware Graph Networks" by Velingker, Sinop et al., the authors show how such measures can be effectively computed in linearithmic time complexity.

Q3. While I find the use of GCNs sensible in the scope of this project, especially if input embeddings Z are more expressive, there could be added benefits to more complex forms of message passing. Specifically, I would have at least expected ablating three models (following the "flavour" categorisation of the Geometric Deep Learning proto-book (Bronstein, Bruna, Cohen and Veličković, 2021)):

- * a convolutional GNN (such as the GCN).
- * an attentional GNN (such as the GAT).
- * a message-passing GNN (such as the Graph Network model from Battaglia et al.).

I think adding these ablations (in terms of running time and energy) for the following models would be very instructive for future practitioners using the method, no matter which of them ends up the most performant.

Q4. I find it slightly unusual that the 'DeepFDL' baseline is named as such, even though the projection to 3-dimensional targets is done by a single affine layer. Therefore the model is, in neural network terms, actually shallow. Have the authors attempted using deeper variants here?

Q5. Because one of the key talking points of the method is its scalability to large graphs, and the authors have identified the GNN as a possible computational bottleneck in this regard, the authors might also find it useful to discuss the various useful works on scaling up GNNs. In the very least, I would mention GraphSAGE (Hamilton et al.) and ClusterGCN (Chiang et al.), discussing to what extent would they be applicable in this pipeline.

Q6. Minor: the current text reads: " L is the graph Laplacian with time complexity $O(N)$ for sparse graphs" -- L , being a matrix, is not an object that has a time complexity. Perhaps the authors meant 'space complexity'?

Q7. Minor typos:

Figure 1d), caption reads "The evolution of a simple cubic lattice, starting from a random configurations"  should be 'configuration';
Figure 2a), caption reads "Comparing the training steps of FDL (left column) and

NeuraLayout (right column)"  Why not use 'NeuLay' here, as was used throughout the document?

I trust the authors should be able to address all of my comments in a response. Provided my concerns are appropriately addressed, I think this will be a very strong paper.

Response to the Reviewers

Comments by Reviewer #1

This paper studies a very interesting and important question: how to use graph neural networks to accelerate network visualization layout. This is timely work. We have more data about large-scale networks than ever before. However, we can only visualize a part of the original network using different sampling methods, preventing us from capturing the whole picture of the entire network. In this work, the authors show that deep learning can address both limitations of the FDL algorithm. Overall, the paper is well written. Yet, I do have several concerns and comments on the current form of the manuscript regarding the models, methods, and results of this work.

We thank the Referee for the excellent summary.

1) It is unclear what are the meanings of the symbols in Fig 1. (g, h, i, j). One can use different function forms for the same data points to fit. Why did the authors choose these?

The distinct symbols present different graph generative models, such as square Random Geometric Graphs (RGG), diamond Barabási-Albert (BA) model, and dots for Erdős-Rényi (ER) random graph algorithm. It is true that different functions can fit the data points. We chose the simplest power-law fit, which seems to do a reasonable job of approximating the data. Our analytical results for the speedup only predict a correlation with the number of outlier eigenvalues, and not predict the functional form, hence the fit is only a guide for the eye.

2) Figure 3a indicates that the loss energy of FDL is much larger than NeuLay when Time is small. It seems that the initial loss energy of FDL is very large. If they start from similar initial values, maybe their performances are identical. Furthermore, the curve of NeuLay is less stable than FDL. Here the authors only show 5 runs. Perhaps it is better to show the results for more runs and include the error bar.

We initialize both FDL and NeuLay with the same Gaussian distribution. However, due to the GNN reparametrization in NeuLay, the initial positions (output of GNN) automatically separate the clusters. Hence, the energy of the layout starts from lower values in NeuLay, despite having the same random gaussian values for all parameters. The energy also drops rapidly in the early stage, hence it looks like we start from different initialization. The unstable nature of the loss curve is caused by the short-range repulsion force among the nodes.

We wish to thank the Referee for the recommendation, and we changed Fig. 3 to show the mean with error bars.

3) The authors focus more on the runtime of the approach but do not discuss the performance enough. Therefore, although we observe some useful visualizations from the illustrative figures (Figures 1d and 2a), I am still unsure about the "performance." Otherwise, it is unclear whether the visualization is similar to or even better than the approaches you compared.

The "performance" of a layout algorithm is assessed through two parameters: the final energy and the time to get to that energy. In the revised manuscript, however, we have ignored our discussion on how do we measure the quality of the layout, and we discuss in detail two different metrics, link length distribution, and similarity ratio to show the performance of NeuLay in finding the communities (See in the SI. Fig. S7, S8).

In summary, we wish to thank Referee 1 for the strong endorsement of the manuscript and for the constructive recommendations.

Comments by Reviewer #2

The paper presents a new computational approach to generating network layouts. Instead of using, say, spring-like layout algorithms, the paper suggests a machine learning method based on the use of graph neural networks. The main goal is to speed up the computation of the layout structure. A secondary goal is to improve the quality of the layout. The paper is well presented and with good illustrations of how this would work in large networks.

We thank the Referee for the succinct summary of our main findings.

1) I am not sure that the main limitation of existing network layout algorithms is the computation cost for large networks. Network layouts are useful for visualization (e.g., to generate an image of the network that can help interpretation of specific results). These images are most useful when the network is small enough for the individual nodes and links to be visible. For very large networks, the network layout is less informative. The computational cost of existing methods is still reasonably low. Force-directed layout algorithms, which have a cost of N^2 , can be used in quite large networks.

We fully agree that most researchers use layout algorithms for small networks, and for that purpose variants of FDL as implemented by visualization software, do a reasonable job. Yet, there is a real need to laying out larger networks, and one of the reason we do not see more large networks visualized is because we do not have good, fast and easy to use tools for it, hence the outcome is not informative. We have experienced this firsthand when we worked with *Nature* to lay out the 80,000 node size citation network for the 150 years anniversary of the journal, and we had to develop special tools for the final layout. We continue to get multiple requests for the tool even now, three years later, indicating a real unfilled need

among researchers to explore large networks. The Referee also points to an important issue: layouts for large networks maybe less informative. Our experience shows that, many large networks have informative large-scale structures that remain hidden if our layout algorithms is not able to extract their main structural characteristics and find a way to display them. This is precisely what we find in the case the Internet map. Although FDL may be reasonably efficient in laying it out, it eventually returns a hairball, independent of the running time (Fig 3). Thus, one might conclude a layout for the Internet network is not informative. Yet, we show in this paper that FDL systematically fails to find a low-energy layout for very large networks. In contrast, NeuLay produces a high-quality layout, with distinct clusters and a clear large-scale structure. Hence, we fully agree that while FDL may be relatively low cost, but it is prone to get stuck in local minima or saddle points for long times. Our algorithm seems to significantly improve on such cases. It also shows that, powerful algorithms like NeuLay may reveal structures in large networks which we did not know. We want to thank to the Referee for bringing our attention to this issue, and we have added a discussion about this to the paper in the Discussion section.

2) Given the above, the main contribution of this paper might is the possibility of a GNN algorithm that can generate better (rather than faster) layouts. This is briefly discussed in the paper, but I wish this could be further developed and really made into the main point (to me, the computation time comes across as a secondary benefit).

We fully agree that the layout quality is a key contribution and have edited the text to emphasize this point further. We introduce two metrics to assess the quality of layouts: 1) cluster separation, and 2) link length distribution in the network (See in the SI. Fig. S7, S8). For large real-world networks we find that NeuLay separates network clusters much better and identifies a much narrower link-length distribution than FDL. Prompted by the Referee's recommendation, we changed the main text and emphasized our efforts and metrics to explore the quality of the layout and discuss explicitly that NeuLay indeed generates a better or at least as good layouts than FDL.

3) A problem with GNN (and any machine learning approach) is that the result will depend on the model training. It is not entirely clear the best way to do that here. Some in-depth discussion seems necessary.

As multiple reviewers have asked about this, it is clear that our formulation of the training has left room for a misunderstanding. To clarify, NeuLay is an *unsupervised* machine learning model, meaning that it does not require a separate training data as supervised machine learning models. Hence we do not train the GNN once and then find layouts using a trained GNN. Rather, the layout X is encoded in the GNN weights θ , reparametrized as $X(\theta)$. A key feature of GNN is that it is not trained separately. Finding the layout means minimizing the loss for the GNN model. Hence, finding the layout *is the training itself*. To find a new layout, the whole GNN is trained again. We have now made this explicit in the *Neural Networks for graph layout* section.

4) Another thing that is not sufficiently clear is the loss function being optimized. Minimizing energies is a natural choice in traditional layout algorithms based on forces, but is that really the best function to optimize in a machine learning implementation? But if the same function is used, then improvements in layout quality will be limited by that choice.

Note that in NeuLay, we are solving the same optimization problem as FDL attempts to solve: finding the node positions which minimize the FDL loss. What the GNN reparametrization means is that we look for minima that can be expressed as the GNN output $w(\theta, z)$. Choosing a different loss function would mean that we are no longer solving FDL.

5) More broadly, if we agree that energy is not necessarily the best function, how to compare the quality of the layouts generated using the GNN algorithm with the layouts generated using FDL algorithms?

We continue to be convinced that energy is the best quantitative way to assess the quality of the layout, and we do find that it correlates with various metrics such as the link length, the linking number [1], and link crossings [2]. Nevertheless, we agree with the Referee that we need additional metrics of quality, to serve as independent measures. For this reason, in Fig. 3 we explore the distribution of link lengths as a measure of quality, showing NeuLay’s ability to find layouts with a narrower link distribution, meaning that connected nodes are closer in the final layout than in the one identified by FDL. Additionally, we introduced a spatial similarity ratio quantifying the spatial separation of clusters in the final layout compared to the FDL layout (See in the SI. Fig. S8). Prompted by the Referee’s comment we have added to the main text a discussion on these methods in sections: *NeuLay offers more optimal layouts faster* and *The quality of large network layouts*.

6) Fig 4 might be the best visual comparison between the methods. By looking at the figure only, it is not immediately clear the advantage given by the new method in terms of layout quality. There might be a way to improve that.

We fully agree, just looking at the layout as an image, we get a limited view. One would need to inspect it from multiple closeups to truly appreciate the difference which is there, however. For this reason, in Fig. 3 we use additional measures, like the energy and the link length distribution to quantify the difference between the two layouts. Also, we introduce a spatial similarity metric to measure how well are the clusters separated from each other (See in the SI. Fig. S7, S8). Finally, we created a 3D video, to view the layout from multiple angles, see <https://vimeo.com/732791412>.

Minor: Why is it said that the DeepFDL architecture maps to the target dimension $d = 3$? Isn’t the layout in 2D?

How do the authors plan to make the codes available to the readers?

The algorithm can generate layouts in any number of dimensions, including 2D and 3D. However, our focus here is on 3D layouts. This is visible again on the video of the Internet

network 3D layout: <https://vimeo.com/732791412>.

The code is available on github as a python package (See on the link: <https://github.com/csabath95/NeuLay.git>).

Finally, we would like to express our sincere gratitude to Referee 2 for the detailed and invaluable comments. The comments have led to vast improvements of our paper and a deeper understanding of the problem.

Comments by Reviewer #3

In this paper, the authors tackle the highly important task of network visualisation. Being able to visualise graphs well is a critical open problem, which usually forms an integral part of any exploratory phase of network science projects. The quality of the visualisation can often make a significant impact on the salient patterns detected in the data. Hence it is an important and timely problem to solve, certainly within scope for Nature Communications.

The authors propose to plug a graph neural network inside the pipeline that produces node 3D coordinates, which are then optimised using a force-field method (energy minimisation with gradient descent).

On the positive side, this work shows clear quantitative and qualitative outperformance compared to the previous state of the art, demonstrating that GNNs can be an excellent tool in the network layout pipeline. I also appreciated the analysis on how this effect likely comes from modelling outlier eigenvalues of the graph Laplacian. Generally, I find the idea to be sound and to make sense, and it produces network layouts of clearly superior quality, especially for large-scale networks of real-world interest. I particularly liked Figure 4 in this regard -- really beautiful!

However, I do have several questions and concerns which would need to all be clearly addressed before I would be comfortable recommending this work for acceptance. I provide my points here, in rough order of significance.

We are delighted that the Referee found our work enjoyable, and we wish to thank for the

clear summary of our main findings, and the constructive comments below.

Q1. The evaluation of the paper is described as follows: "We rely on two figures of merit to assess performance: the running time (number of iterations), that captures the computational cost, and the potential energy (loss value) of the final layout, that measures the quality and ultimately the interpretability of the layout." In my opinion, there are a two concerning / unclear aspects with this description.

a) First, 'running time' typically refers to 'wall-clock time' (in seconds), but the paragraph implies that we're actually counting the number of gradient descent steps. I think making this distinction clearer is quite necessary. Further, for a downstream practitioner, I think wall-clock time, and/or the hardware setup, might also be quite meaningful to report -- especially because the various baselines may benefit differently from GPU accelerators, and practitioners may appreciate these considerations.

b) Secondly, without thorough user studies, I would be *very* careful claiming that the layout's energy is a proxy for its interpretability. Surely, the produced examples in the paper are good in this regard, but I would tone down the certainty of the claim that the energy is a measure of interpretability.

a) We measure the running time in wall-clock time in 'seconds' in all our experiments. NeuLay significantly reduces the iteration steps of the converge. Also, despite the higher cost per iteration of using GNN, NeuLay needs a shorter time to reach the same energy level compared to the FDL. To avoid misunderstanding, we attach loss curves showing the running time and iteration steps in the SI (Fig. S6).

b) We fully agree that what we called interpretability of the layout can be somewhat subjective and may be confusing. It fails to capture the experience we have developed during the project, that did show over and over that energy is a true proxy of the layout quality.

But we agree with the Reviewer that more is needed. We have therefore developed additional measures for the clarity of the layout, such as how well clusters are separated or the link length distribution (See in the SI. Fig. S7, S8). In the context of this paper, we use ‘interpretability’ to refer to the quality of the layout in terms of its ability to capture the network structure, for example, clusters.

Q2. Both the DeepFDL and NeuLay methods start with an embedding vector in each of the nodes, Z . However, after several passes through the paper, I found no clear description in the paper on where this Z comes from. My assumption is that it could be a standard structural node embedding method (e.g. DeepWalk, or the lowest h Laplacian eigenvectors). In either case, it is critical to describe where Z could come from, and also to discuss the computational complexity of obtaining such Z (since, e.g., computing Laplacian eigenvectors may be intractable for too large graphs). On a related note, the authors may want to consider the use of affinity-based measures (e.g. effective resistance or hitting time) for powerful embeddings. In the recent preprint "Affinity-Aware Graph Networks" by Vellingker, Sinop et al., the authors show how such measures can be effectively computed in linearithmic time complexity.

Thank you for pointing out the need for clarification of Z . This is an important, yet subtle point. The key to the point is that FDL directly optimizes X , while NeuLay optimizes $X(Z, W, b)$ *indirectly* by optimizing Z and GNN parameters W, b , instead. For *each* layout Z is *initialized randomly*. Z is then optimized together with the GNN parameters W, b . Hence, there is no extra step to run to find the initial node embedding method. Z is similar to the latent space of the “generator” in a GAN (except that Z is trainable), and different from a DeepWalk embedding.

For each node, i , we have a random latent vector Z_i , that describes a random position in a high dimensional latent space. Z is easier to understand in the DeepFDL case. There,

the MLP layers are only doing a nonlinear projection of the $Z_i \in \mathbb{R}^h$ to the 3D positions $x_i \in \mathbb{R}^3$. Hence, in DeepFDL, instead of optimizing node positions in 3D, we optimize their high dimensional latent positions, which is Z . In DeepFDL, we are also finding the best projection down to 3D (the FC layers). In NeuLay, Z_i is not directly interpretable as the latent embedding for node i , because the message passing mixes Z_j from neighbors of each node i to yield the final position x_i .

Q3. While I find the use of GCNs sensible in the scope of this project, especially if input embeddings Z are more expressive, there could be added benefits to more complex forms of message passing. Specifically, I would have at least expected ablating three models (following the "flavour" categorisation of the Geometric Deep Learning proto-book (Bronstein, Bruna, Cohen and Veličković, 2021)):

- * a convolutional GNN (such as the GCN).
- * an attentional GNN (such as the GAT).
- * a message-passing GNN (such as the Graph Network model from Battaglia et al.).

I think adding these ablations (in terms of running time and energy) for the following models would be very instructive for future practitioners using the method, no matter which of them ends up the most performant.

We thank the Referee for the excellent recommendations. We have followed them and reimplemented the code using Pytorch Geometric library and rerun all experiments using GCNConv [3], GATConv [4] and Graph Network (GN) [5] models. Results from GCNConv and GN show a slightly better time and energy performance than GAT. Also, all models show significant speedup improvements compare to the baseline FDL model. We have now added these results to the manuscript at the SI (Fig. S11.).

Q4. I find it slightly unusual that the 'DeepFDL' baseline is named as such, even though the projection to 3-dimensional targets is done

by a single affine layer. Therefore the model is, in neural network terms, actually shallow. Have the authors attempted using deeper variants here?

We originally named the model DeepFDL because, in principle, it can consist of a deep MLP, acting node-wise. Yet, we agree with the Referee’s pertinent observation and we have changed the DeepFDL name to NodeMLP. We did test deeper neural networks, but we couldn’t achieve speedup for large graphs because of the computational cost. Also, the hyperparameter tuning of the NodeMLP model turns out more difficult, and the model suffers from finding a good layout for large networks.

Q5. Because one of the key talking points of the method is its scalability to large graphs, and the authors have identified the GNN as a possible computational bottleneck in this regard, the authors might also find it useful to discuss the various useful works on scaling up GNNs. In the very least, I would mention GraphSAGE (Hamilton et al.) and ClusterGCN (Chiang et al.), discussing to what extent would they be applicable in this pipeline.

Thank you for the suggestion. Indeed, using GNN for large dense network reparametrization is expensive. As a future step, we will try to utilize the idea behind GraphSage [6] and ClusterGCN [7] for the reparametrization. We have tried to apply a low-rank adjacency matrix, using the largest eigenvalue eigenvectors, in the propagation rule, and we see some improvements. However, it needs further analysis. In the same way, making the adjacency matrix sparser by randomly removing neighbors for each node or keeping the clusters together and removing interconnected edges could solve the model’s scalability limitation. We should note that each of these methods makes assumptions about what property of the graph we should preserve during sparsification (e.g. cluster structure or eigenvectors), which requires justification in our problem. We have added a discussion in the Discussion section.

Q6. Minor: the current text reads: "L is the graph Laplacian with time complexity $O(N)$ for sparse graphs" -- L, being a matrix, is not an object that has a time complexity. Perhaps the authors meant 'space complexity'?

Q7. Minor typos:

Figure 1d), caption reads "The evolution of a simple cubic lattice, starting from a random configurations"  should be 'configuration'; Figure 2a), caption reads "Comparing the training steps of FDL (left column) and NeuraLayout (right column)"  Why not use 'NeuLay' here, as was used throughout the document?

We pre-compute the Laplacian matrix, denoted as L . The time complexity for computing the elastic-energy term of L , denoted as LX , is $O(N)$ for sparse graphs and $O(N^2)$ for dense graphs, where N is the number of nodes. We term a graph as sparse if the number of edges grows linearly with nodes, that is, $O(n) \leq edges \leq O(n^2)$.

We corrected both typos.

In summary, we wish to thank Referee 3 for the pertinent questions about the relevance of our methodology and the significance of our work. Addressing these issues has contributed significantly to the conceptual contributions of our paper.

References

1. Liu, Y., Dehmamy, N. & Barabási, A.-L. Isotopy and energy of physical networks. *Nature Physics* **17**, 216–222 (2021).
2. Kobourov, S. G., Pupyrev, S. & Saket, B. *Are crossings important for drawing large graphs?* in *International Symposium on Graph Drawing* (2014), 234–245.
3. Kipf, T. N. & Welling, M. Semi-supervised classification with graph convolutional networks. *arXiv preprint arXiv:1609.02907* (2016).

4. Veličković, P., Cucurull, G., Casanova, A., Romero, A., Lio, P. & Bengio, Y. Graph attention networks. *arXiv preprint arXiv:1710.10903* (2017).
5. Battaglia, P. W., Hamrick, J. B., Bapst, V., Sanchez-Gonzalez, A., Zambaldi, V., Malinowski, M., Tacchetti, A., Raposo, D., Santoro, A., Faulkner, R., *et al.* Relational inductive biases, deep learning, and graph networks. *arXiv preprint arXiv:1806.01261* (2018).
6. Hamilton, W., Ying, Z. & Leskovec, J. Inductive representation learning on large graphs. *Advances in neural information processing systems* **30** (2017).
7. Chiang, W.-L., Liu, X., Si, S., Li, Y., Bengio, S. & Hsieh, C.-J. *Cluster-gcn: An efficient algorithm for training deep and large graph convolutional networks* in *Proceedings of the 25th ACM SIGKDD international conference on knowledge discovery & data mining* (2019), 257–266.

REVIEWERS' COMMENTS

Reviewer #1 (Remarks to the Author):

The authors have addressed all my comments, and the manuscript is ready for publication.

Reviewer #2 (Remarks to the Author):

The authors have given proper attention to the points I raised and revised the manuscript accordingly. I support publication of the current manuscript.

Reviewer #3 (Remarks to the Author):

I would like to thank the authors for carefully responding to all of my points. I am fully satisfied by the changes, and wholeheartedly support this work for acceptance. Great work!

REVIEWERS' COMMENTS

Reviewer #1:

The authors have addressed all my comments, and the manuscript is ready for publication.

Reviewer #2:

The authors have given proper attention to the points I raised and revised the manuscript accordingly. I support publication of the current manuscript.

Reviewer #3:

I would like to thank the authors for carefully responding to all of my points. I am fully satisfied by the changes, and wholeheartedly support this work for acceptance. Great work!

We would like to express our sincere gratitude to all Referees for the detailed, invaluable, and very enlightening comments. The comments have led to vast improvements of our paper and a deeper understanding of the problem.